# Rainwater Catchment System Reliability Analysis for Al Abila Dam in Iraq's Western Desert

Ammar Adham [1,2,*], Rasha Abed [2,3], Karrar Mahdi [2], Waqed H. Hassan [4,5], Michel Riksen [2] and Coen Ritsema [2]

1   Dams and Water Resources Engineering Department, College of Engineering, University of Anbar, Baghdad 55431, Iraq
2   Soil Physics and Land Management Group, Wageningen University, 6700 AA Wageningen, The Netherlands
3   Anbar Technical Institute, Middle Technical University, Baghdad 10074, Iraq
4   College of Engineering, University of Warith Al-Anbiyaa, Kerbala 56001, Iraq
5   College of Engineering, University of Kerbala, Kerbala 56001, Iraq
*   Correspondence: engammar2000@uoanbar.edu.iq

**Abstract:** Rainwater Catchment System Reliability (RCSR) is the chance that a system will deliver the required water for an interval of time. Rainwater Harvesting (RWH) is gaining popularity as a potential alternative water source for household or agricultural use. The reliability of the Al Abila dam in the western desert of Iraq was analyzed using a water budget simulation model and two explanations of reliability, time-based reliability, and volumetric reliability. To evaluate rainwater harvesting system performance, comprehensive software utilizing a method for everyday water balance using data from 20 years of daily rainfall. According to the findings, volumetric reliability, and for the three climate scenarios (wet, average, and dry year), increased as the storage volume increased until a threshold accrued on the storage capacity of $11.7 \times 10^5$ m$^3$. While time-based reliability shows an increase up to a storage volume of $10.2 \times 10^5$ m$^3$. Volumetric reliability of roughly 34–75% may be achieved, while only 14–28% time-based reliability may be achieved. Water saving efficiency decreases with increasing demand fraction, while the runoff coefficient has no significant influence on water effectiveness. While growing storage fraction value increases the effectiveness of water conservation and the value of the runoff coefficient influences the water saving efficiency. For both cases, water saving efficiency for the dam does not reach 50%. Using daily rainfall data, the technique given in this paper might be applied to predict water savings and the RWH systems' reliability in different arid and semi-arid areas.

**Keywords:** reliability; rainwater catchment; Al-Abila dam; Iraq

## 1. Introduction

Iraq was seen as having abundant water supplies until the 1970s. However, water shortages in Iraq have been brought on by the building of dams in the Tigris and Euphrates Rivers and their tributaries outside the Iraqi border, as well as by increasing water consumption, population growth, and urban and industrial expansion. Therefore, water scarcity is among the most serious issues in arid and semi-arid regions, especially in developing areas such as the western desert of Iraq [1,2]. As a result, both developing policies and technology to locate alternate water supplies, and enhancing water resource management and planning, will be crucial. Rainwater Harvesting (RWH) systems are gaining popularity as an alternative water supply, and are seen to be viable approaches for storing water for home or farming purposes [3–5]. However, little is known about their effect on the chance that these systems provide the required water for a period of time, here defined as the Rainwater Catchment System Reliability (RCSR)A water balance model based on input and output flow may be used to evaluate available water supply [6–8]. The water balance modeling is also useful for calculating RWH reliability. Determining the reliability of an

RWH system is a key aspect to assess demand reliability, the probability that the system will satisfy the water demand for a certain timeframe is described as reliability [9–12].

Baek and Cole [9] evaluated the impact of the modeling time period for the water balance models and the concept of reliability to determine the variation in the watershed systems' reliability. Five concepts of reliability and weekly and daily modeling time periods are used to assess the reliability of catchment dam systems at ten sites in Western Australia's dryland rural regions.

According to evaluation findings, the possibility of underestimation makes utilizing yearly period-based prediction for reliability unsuitable for dry and semi-arid regions of Western Australia. Volume-based predictions as well as weekly and daily period-based predictions have the possibility of being overestimated when the pattern of growing crops and water requirements in Western Australia is taken into account. For the design of water harvesting schemes in the dryland agricultural lands of the southwest of Western Australia, it is therefore advised to employ monthly period-based prediction.

Jafarzadeh et al. [10] evaluated the future reliability of RWH. Using the outcomes from General Circulation Models (GCMs) for both historical and future eras, monthly rainfall was predicted in the first stage. Data was then spatially downscaled. For each month rainfall was interpolated for future periods using the standard kriging approach. Finally, the reliability of RWH was evaluated and investigated for various roof areas and storage tank capacities. Findings demonstrate that a reliability band of 0.05–0.45 RWHS was calculated for the historical period and that this reliability range will increase for the future period based on the best GCMs. Additionally, a variance in RWH reliability revealed that, in general, RWH reliability under Representative Concentration Pathway (RCP), 2.6 rcp will be greater than 8.6 rcp in the future.

Imteaz et al. [13] created a tank tool using daily water balance simulation. In several Australian cities, including Melbourne, the advanced approach was regularly employed to assess RWH tank's reliability. Researchers looked at how reliable a specific magnitude of rain water tank is in relation to annual volume and meeting daily estimated requirements in Bangladesh's megacity [14]. In conjunction with the town water delivery systems in Dhaka City, this article examines the economic viability, adaptability, and reliability of rainwater harvesting (RWH) systems to partially balance the daily water requirements in multistory buildings. To evaluate the reliability and viability of the RWH systems in an urban setting, extensive computer program was created. By examining daily rainfall data for the previous 20 years, three distinct climate scenarios—rainy, normal, and dry years—were chosen. Results showed that within the wet climatic condition, roughly 15–25% reliability may be reached [14].

Male and Kennedy [15] investigated the probable role of rainwater usage for home uses in Portland, Oregon, with a focus on rainwater collection reliability. Applying the water balance, they detailed the technique using the amount of rainfall collected, domestic demand, and capacity of storage tanks. The capacity of the storage tank, in addition to the catchment area's size, was shown to be essential in determining the system's reliability.

The reliability of RWH is critical to residents' desire to know that it might be necessary for their source of water [15,16]. Nevertheless, no comprehensive research has been undertaken as yet on the feasibility and RWH reliability collection technology in the sub-catchment [17,18].

Liuzzo et al. [19] looked at the efficiency of a potential RWH tank for a model single-family home in a neighborhood. Information from more than 100 locations in Sicily was used to test performance for various yearly precipitation amounts. The performance was evaluated for three uses of the rainwater collected and three storage sizes (10, 15, and 20 m$^3$). The system's reliability was examined as a function of average annual rainfall after the system's performance for the full research area had been assessed. This analysis allowed for the development of mathematical equations with regional applicability and implementation. To determine the degree of uncertainty surrounding the regional model provisions, a data resampling approach was used. To determine the payback period for the

capital cost associated with the installation of the RWH system, a cost-benefit analysis was lastly carried out.

In contrast, in a developing country such as Iraq, specifically in the western desert, where the technique for supplying water is below enormous stress and can be vanished, there are few comprehensive studies on the possibility of harvesting rainwater. In addition, there is no in-depth research about the sustainability, reliability, or efficiency of any planned RWH systems in the area, the studies are limited to investigating the suitability of the sites and the techniques to implement RWH.

The major goal of this study is to evaluate RWH systems reliability in a sub-catchment applying a water balance technique, by examen how much the reliability of RWH in the sub-catchment of the Al Abila dam is influenced by the storage capacity. This will be done by estimating the volumetric reliability and time-based reliability for wet, average, and dry-year climate scenarios.

## 2. Methodology

### 2.1. Area and Data Utilised

Wadi Horan is placed in Al-Anbar region in western Iraq, around 450 km west of Baghdad (Figure 1). The watershed is about 370 km$^2$ in size and has a dry environment by dry summers and mild winters. The average yearly rainfall is just 115 mm, where the winter months get around 49% of the rain, the spring months 36%, the fall 15%, and the summer months see no rain. The average annual temperature is 21 degrees Celsius, with July being the warmest month and January being the coldest [3,20]. The yearly evaporation possible is 3200 mm on average.

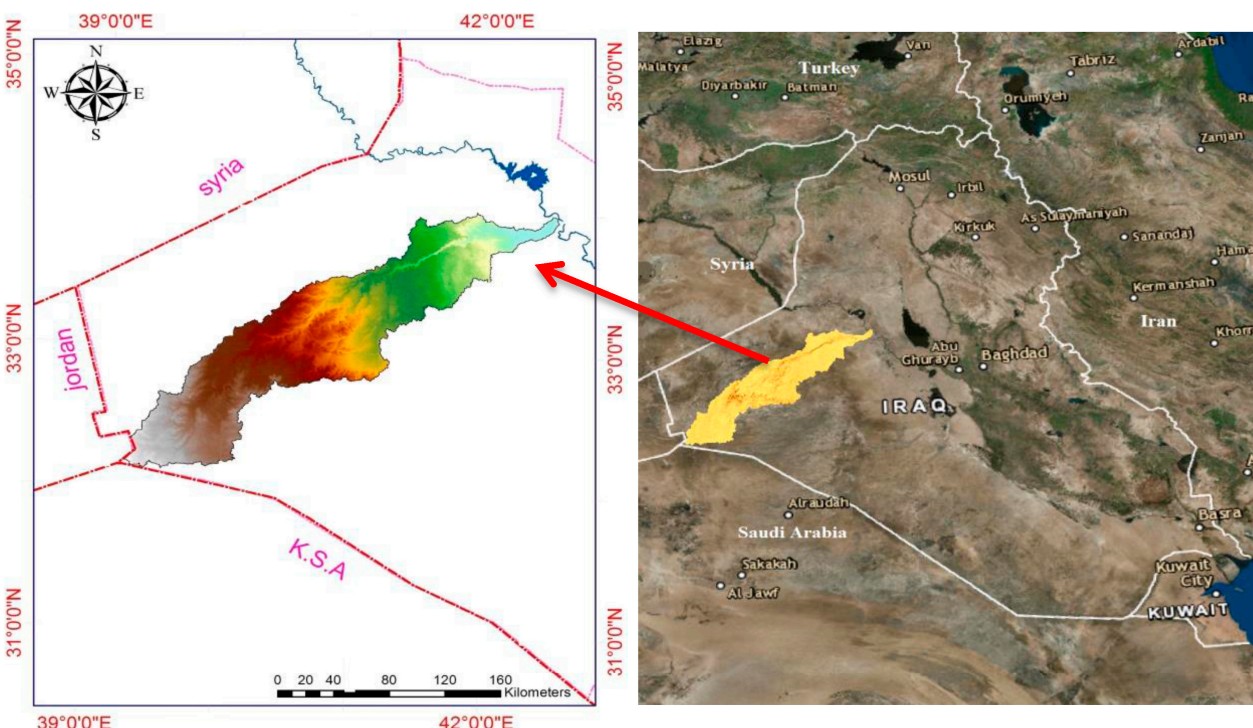

**Figure 1.** Study area, showing the Al-Abila dam location after [3].

Hard limestone makes up the majority of the wadi Horan's exposed rocks [20]. They can be utilized to protect the front edge of barriers and make a nice foundation for dams or other barriers. In order to minimize the number of building materials required for the dams, reduce evaporation losses, and guarantee the necessary storage, the catchment area and potential for a hard, narrow cross-section of the wadi with vertical shoulders were taken into con-sideration while deciding where to place the dams.

To test the RCSR of the Al Abila dam, a 30-year rainfall dataset (1990–2020) was used to determine the three distinct meteorological years: wet (max. rainfall), average, and dry (min. rainfall). An average year was defined as one equivalent to a regular yearly rainfall over a period of approximately 20 years.

The physical characteristics of the Al Abila catchment were assessed. Using a tape measure and the Global Positioning System (GPS), the height of the dike and spillways for the Abila dam were determined. These data were used to determine the entire volume of water that might be gathered behind the dam. In the Abila catchment, the texture of soil was evaluated by aggregating samples, and the slope and area were observed in the field. DEMs and a Geographic Information System (GIS) were used.

A rainfall simulator was used to quantify runoff coefficients at many places in the Al Abila catchment. A Kamphorst, [21], rainfall simulator was used to simulate rainfall in the Abila catchments' area. A device called a "rainfall simulator" attempt to replicate the physical traits of natural rainfall as accurately as possible [22]. The instrument was calibrated in line with Kamphorst's instructions (1987). Each test took three minutes to monitor the water level, taking readings every 30 s. A tube was used to catch any runoff, and the amount was measured. At the completion of each simulation, the runoff coefficient (C) value was computed.

The infiltration rates of the Al Abila dam were evaluated. The infiltration rate was determined using a double-ring infiltrometer [23]. We made use of infiltrometers with 18/30 cm internal and external rings. In order to assure a trustworthy result, tests were often conducted twice for each site. Preliminary filling of internal and external rings was to a depth of 15 cm. More water was given to maintain equal levels once the water level in the external ring dipped just below the level in the internal ring. A scale mounted on the internal ring was used to measure the water level as a function of time during the test. We kept doing this until the level of water fell to less than 5 cm, the water was then supplied for the following iteration. In most cases, one to four repeats were carried out to guarantee that a steady infiltration rate was attained. These results were used to assess the Al Abila catchment's average infiltration rate over a specific time period. The Water Harvesting model (WHCatch) [24] uses these data types as input.

*2.2. Water Harvesting Model (WHCatch)*

To analys the effectiveness of the RWH approaches based on current climatic circumstances, we used the WHCatch basic model [24] for the Al Abila dam watershed. Based on the water demands, the water supply, and the structures design of the RWH, the water balance of the Al Abila dam was examined. The variance between the total input and output was used to compute the variation in water storage volume. A runoff area and a reservoir area are the two basic components of a catchment. To increase the RWH system's reliability, we evaluated the performance of RWH throughout the entire system and examined the water balance of these two components. The water storage change in the Al Abila dam was calculated by subtracting the entire inflow from the total outflow [25]:

[Water balance calculation of any location in m³:]

$$\Delta S = I - Q \qquad (1)$$

where $\Delta S$ is the storage change during a certain time period, $I$ is the input flow, and $Q$ is the output flow, all together in m³. The details of this model (WH Catch) and its application with the manual were explained and published in two articles [24]. In MS Excel, the Al-Abila dam's monthly water balance study was carried out to assess the dam's effectiveness in satisfying local water demands. We used the WHcatch program, a straightforward Visual Basic for Applications (VBA) macro in Excel, since all input data were logged and available there. The computations were carried out by this macro, which then recorded the results in the appropriate cells. A WHCatch module and a Sub-catchment Class module made up the code. The last one included a routine to carry out certain fundamental calculations as well as all the attributes of a sub-catchment. Three general and a few private subroutines made

up the WHCatch module. The VBA macro won't be visible to the Excel workbook's regular users. Only when the further capability is needed, entering the code section will be crucial. This tool allows for the reading of data into GIS applications, as well as all outcome is kept and shown in the same Excel workbook. Nearly all circumstances, the shape file containing the location's layout and the IDs of its sub-catchments are accessible. Thus, the ID in the Excel workbook (column name) and the ID in the sub-catchment ID in the shape file can be collective.

### 2.3. The SCS–CN Method

The Soil Conservation Service (SCS) technique, created in the USA by Soil Conservation Service (SCS) in 1969, is a straightforward, dependable, and consistent intellectual approach for the determination of runoff based on rainfall. It simply depends on the variable CN. We calculated the runoff depth using the Curve Number (CN). Following that, the depth of runoff is applied to calculate the probable water supply following runoff. The influence of soil and land use on precipitation and runoff makes CN dependable. An expression for runoff depth is:

$$Q = \frac{(P - I_a)^2}{(P - I_a) + S} \tag{2}$$

where $Q$ is the depth of runoff (in mm), $P$ is the amount of rainfall (in mm), $S$ represents the possible maximum retention (in mm), and $I_a$ is the initial abstraction (in mm), which accounts for all losses prior to the start of runoff, infiltration, evaporation, and water interception. The rainfall data for numerous small rural regions were analyzed to arrive at $I_a = 0.2S$.

### 2.4. Reliability Analysis

This research revealed two reliability categories. The following Imteaz equation is used to calculate time-based reliability [13]:

$$R_t = \frac{T_d - U_d}{T_d} \times 100 \tag{3}$$

where $R_t$ refers to time-based reliability (percentage), $U_d$ represents the number of days where RWH was inadequate for daily demand, and $T_d$ is how many days there are in a year (365).

The second type of reliability is volumetric reliability ($R_v$) which is given by:

$$R_v = \frac{\sum (VW_d - VD_d)}{\sum VW_d} \tag{4}$$

where $VD_d$ is the annual water deficit and $VW_d$ is the annual water demand.

### 2.5. Sensitivity Analysis

Sensitivity assessments were carried out to examine the effect of the coefficient of runoff on the efficiency of RWH storage and demand fraction. Four values of runoff coefficient (from 0.3 to 0.6) were considered (C1 = 0.3, C2 = 0.4, C3 = 0.5, C4 = 0.6) taking into account the infiltration and the spilling losses from the rainfall.

The relation of water saving efficiency with demand fraction and storage fraction, respectively, were shown in sensitivity figures. Demand fraction (*D/Q*) is calculated using the formula below:

$$\frac{D}{Q} = \frac{VW_d}{VW_s} \tag{5}$$

where $VW_d$ refers to annual water demand and $VW_s$ refers to the annual volume of rainwater supply.

The storage fraction (*S/Q*) is calculated using the following formula:

$$\frac{S}{Q} = \frac{SW_d}{VW_s} \tag{6}$$

where $SW_d$ is the annual storage capacity of rainwater supply.

## 3. Results and Discussion

The initial evaluation was carried out to see how storage capacity affects daily reliability and to determine the amount of storage that offers the Abila Dam's highest average reliability value. The model of water balance was run on a daily basis, and throughout the investigation, the Abila dam's daily average reliability was calculated. The associated percentile values were then calculated.

### 3.1. The Al Abila Dam Assessment

Based on the greatest depth of daily rainfall measured from the Al-Rutba station for the years (1990–2020), the data was examined using the water balancing approach in the Abila watershed utilizing the Water Harvesting model (WHCatch) under Microsoft Excel to determine the variation in water storage within the volume.

The Abila Dam has a $4 \times 10^6$ m$^3$ design capacity. Only once, in 1994, did the dam's reservoir fill to its intended level, as illustrated in Figure 2. Additionally, there was essentially little runoff from 2000 to 2009, which left the dam's reservoir dry and the dam inoperable.

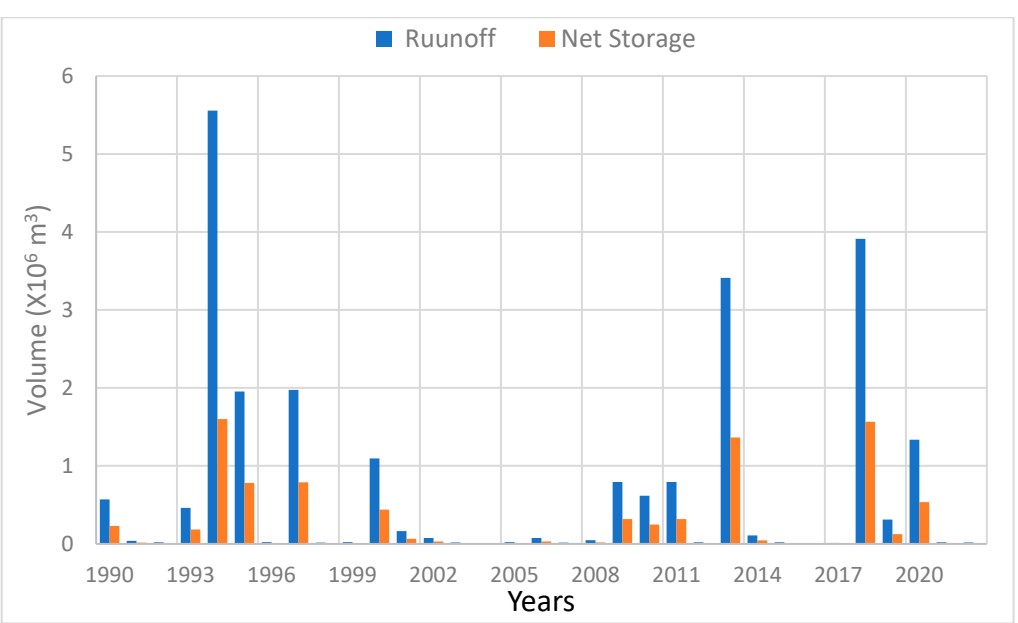

**Figure 2.** The total annual runoff volume and net storage volume.

The analysis showed that out of the total years (1990–2020) where surface runoff occurred, the number of years where it entered the dam reservoir and caused the dam to store all of its intended $4 \times 10^6$ capacity was 5.2%. According to the findings, the dam's reservoir is greater than the available storage. Since the base of the trench has not been deepened to reach the hard rocky layers, the Abila dam has a problem with regular seepage along its body. A back trench (Toe drain) downstream of the dam was also missing.

### 3.2. Reliability Analysis

The relation between the reliability and the storage volume was illustrated in Figure 3, which shows where the reliability increased and then stabilized over the increasing of the

storage volume, for (a) wet, (b) average, and (c) dry year climate scenarios. Each scenario shows similar profiles over the course of the increasing storage volume, while wet scenarios show more reliability than average and dry scenarios due to the additional rainwater in the wet year, which keeps the reservoir full. With the dry-year scenario, reliability was within an acceptance rate as well. With no rainfall period with around 5% to 18% $R_t$ with a storage volume that varies from approximately $4 \times 10^5$ to $10 \times 10^5$ m$^3$ the dam can still provide water.

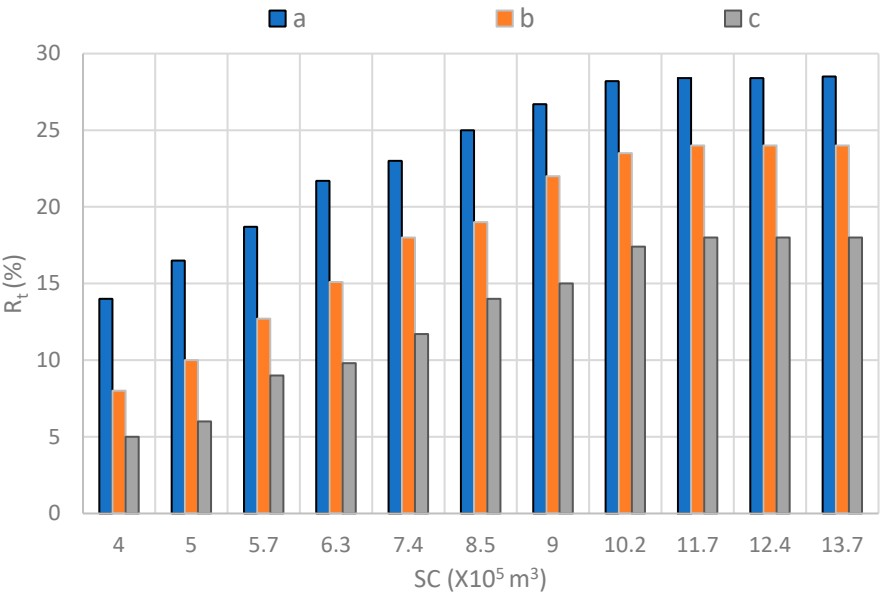

**Figure 3.** The percentage of time-based reliability ($R_t$)vs storage capacity (SC) for (a) wet, (b) average, and (c) dry-year climate scenarios.

The figure also shows that in all three scenarios, a threshold point accrued around a storage volume of $10.2 \times 10^5$ m$^3$. The threshold point specifies the stage at which storage capacity does not affect reliability. In a wet year, if the dam received all the harvested water and its reservoir become full, then increasing the storage beyond this volume is not needed and does not influence the dam's reliability. This also implies the dry-year scenario.

Figure 4 represents the volumetric reliability or the proportion of water saved for a catchment area with varied storage volume capacity. It can be seen that the influence of increasing the storage capacity on volumetric reliability follows a similar pattern as time-based reliability. For wet, average, and dry-year climate scenarios, the volumetric reliability tends to a considerable increase as the storage volume increase until it remains steady around $11.7 \times 10^5$ m$^3$, where increasing the storage volume has no more influence on the reliability of the dam.

For wet and average scenarios, the reliability results were more convergent for the same storage size, for instance: at a storage capacity of $9 \times 10^5$ m$^3$, the percentage of volumetric reliability was 69% and 64% for the wet and average scenarios, respectively. This may mainly be because, according to the calculation, the annual average water demand for the catchment is considered to be constant, and it has the main influence on the value of the volumetric reliability. After then, each distinct scenario's variances in the amount of captured rainfall accumulated.

According to Figures 3 and 4, the volumetric reliability value for the catchment is discovered to be greater than the time-based reliability. Where the time-based reliability varies from 14% to 28%, volumetric reliability ranges from around 34% to 75% for the wet year scenario. This is mostly due to the time-based reliability was computed using the number of days overall during which the rainwater harvest is sufficient to provide the re-quired water demand, and this is relatively not much, due to the local's climatic conditions compared to the amount of water demand required to meet the daily needs.

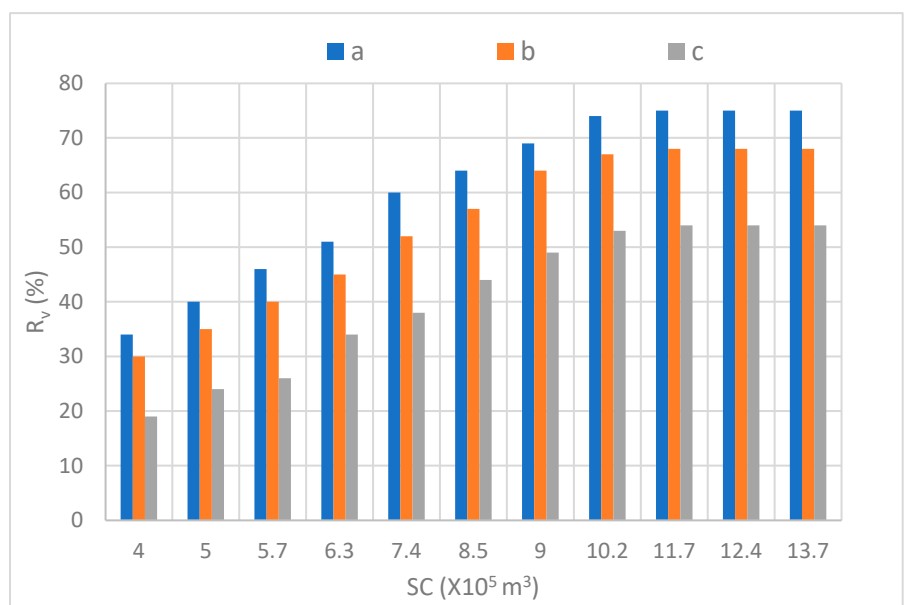

**Figure 4.** The percentage of volumetric reliability ($R_v$) vs. storage capacity (SC) for (a) wet, (b) average, and (c) dry-year climate scenarios.

### 3.3. Sensitivity Analysis

Figure 5 shows how the runoff coefficient (C) affects water-saving efficiency in wet year conditions. Sensitivity analysis was implemented on arrange of runoff coefficient values starting from 0.3 to 0.6. According to the result shown in Figure 5, the coefficient of runoff has no significant influence on water effectiveness in wet climatic conditions, and the variety of C does not imply a significant change in the influence pattern of efficiency as well.

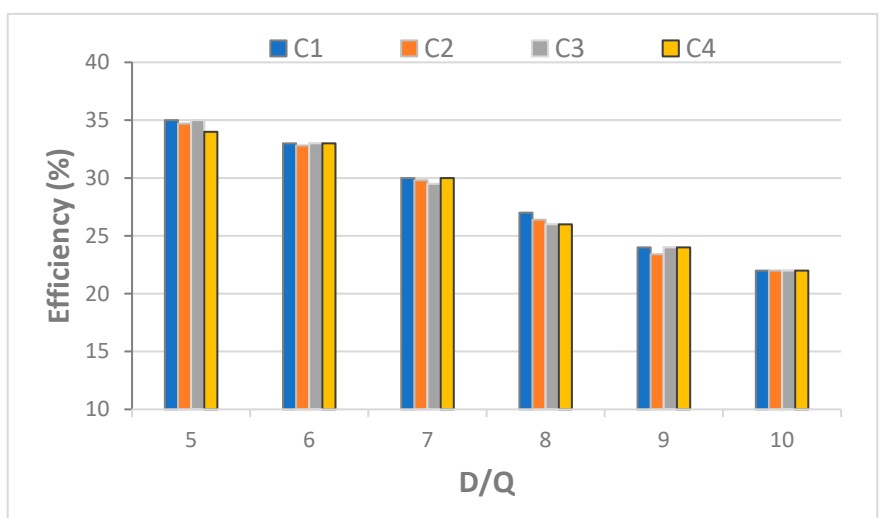

**Figure 5.** Water saving efficiency dealings with runoff coefficients vs. demand fraction (D/Q).

The influence of the coefficient of runoff on water-saving efficiency is seen in Figure 6. The results show that increasing the storage fraction leads to increased efficiency. So, when the *S/D* value increases from 0.07 to 0.09, the efficiency increases by about 2%. In addition, the results show that for the same storage fraction, the value of C has an obvious influence on efficiency. For C1 and C4 there is an average increase of 2% in efficiency.

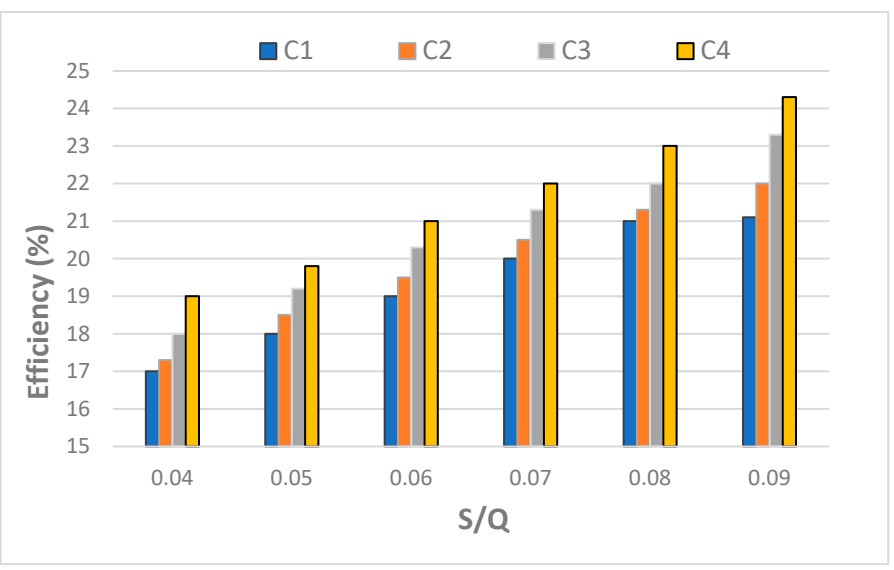

**Figure 6.** Water saving efficiency dealings with runoff coefficients vs. storage fraction (D/Q).

The sensitivity analysis shows that the water-saving efficiency for the dam does not reach 50% for both cases (demand fraction and storage fraction) and for the different C values.

The efficiency has been influenced differently by changing the runoff coefficient as shown in Figures 5 and 6. In the case of demand fraction, the change in the C value has almost no effect on the efficiency within the same demand fraction value. While the efficiency increases slightly with increasing the runoff coefficient for each storage fraction value.

## 4. Conclusions

Recently, awareness of RWH systems as a substitute water supply has grown. These systems can be linked with existing traditional water sources to provide supplemental water supplies in various locations, or they can act as the primary water source in arid and semi-arid regions in which water availability is a major concern. Additionally, using RWH is a successful adaptation technique to combat the decline in water availability caused by climate change. This study evaluates RWH systems reliability in a sub-catchment applying a water balance technique by examen how much the reliability of RWH in the sub-catchment of the Al Abila dam is influenced by the storage capacity. The reliability was investigated using a water balance model, time-based reliability, and volumetric reliability followed by sensitivity analysis. According to reliability correlations with varying storage volumes, time-based reliability shows an increase throughout the wet year, average year, and dry year, equal to a storage volume of $10.2 \times 10^5$ m$^3$, and beyond that, the threshold point accrued when reliability doesn't rise with riding storage volume. The volumetric reliability, and for the three climate scenarios (wet year, average year, and dry year), increased as the storage volume increased until $11.7 \times 10^5$ m$^3$, then increasing the storage volume has no more in-fluence on the reliability of the dam. For the three scenarios, volumetric reliability value for the catchment is detected to be greater than the time-based reliability. The time-based reliability varies from 14% to 28%, while volumetric reliability varies from about 34% to 75% for the wet year scenario. Despite the large storage capacity of the dam, the reliability of the dam does not show significant value.

Water saving efficiency decreases with increasing demand fraction, while the runoff coefficient has no significant influence on water effectiveness in wet climatic conditions. Growing storage fraction value increases the water-saving efficiency, and the value of the runoff coefficient influences the water-saving efficiency. For instance, there is an average

increase of 2% in water efficiency for the same value of storage fraction. For both cases, the water-saving efficiency of the dam does not reach 50%.

For a reliable rainwater harvesting system, this kind of study should be considered as a design guideline to scientifically highlight the system elements that play a significant role in increasing the reliability of any system. For instance, thresholds enable landowners to get the most out of their systems, while saving additional costs of increasing the storage of the rainwater harvesting system (the dam in this case). While the storage capacity is attempting to grow, reliability does not improve. In all situations, volumetric reliability was shown to be greater than time-based reliability, and thus could not be neglected in the design of any new systems. Using daily rainfall data, the technique given in this paper might be applied to forecast water conservation and the RWH systems' reliability in different arid and semi-arid areas.

It is important to include the impact of climate change on rainfall in future analyses. The equations described here hold for both historical and contemporary environmental circumstances. The performance of an RWH system may be considerably impacted by trends. In particular, a significant decline in system efficiency may be caused by a drop in rainfall volume and a variation in the timing of rainfall throughout the year. Deriving future climatic scenarios from regional climate models should therefore be considered while designing the RWH systems.

Finally, understanding the reliability of a rainwater harvesting system is critical, which informs decision-makers and households about the design criteria of the system and how much they expect from the system to meet water demands throughout the year.

**Author Contributions:** Conceptualization A.A. and R.A.; methodology, A.A, R.A. and K.M.; software, A.A. and R.A.; formal analysis, K.M. and M.R.; investigation, A.A., W.H.H. and R.A.; resources, A.A. and R.A.; writing—original draft preparation, A.A., C.R., and R.A.; writing—review and editing, C.R., K.M., W.H.H. and M.R.; visualization, A.A., K.M., W.H.H. and M.R.; supervision, C.R.; project administration, C.R.; funding acquisition, C.R. All authors have read and agreed to the published version of the manuscript.

**Funding:** This research received no external funding.

**Data Availability Statement:** Some data in this manuscript was obtained from the Ministry of Agriculture and the Ministry of Water Resources, Iraq. The other data from the fieldwork and previous studies.

**Acknowledgments:** This study was funded by the NUFFIC Orange Knowledge Program (OKPIRA-104278) and coordinated by Wageningen University & Research, The Netherlands.

**Conflicts of Interest:** The authors declare no conflict of interest.

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
