# Peer review of "Rainwater Catchment System Reliability Analysis for Al Abila Dam in Iraq’s Western Desert"

_water, doi:10.3390/w15050944_

Round 1

Reviewer 1 Report (New Reviewer)

The article is well written, and focused on an actual problem in water management issues for Al Abila Dam in Iraq's Western Desert and provides evidence of the reliability analysis of the dam.

The title is appropriate for the content of the article. The abstract is concise and accurately summarizes the essential information of the paper although it would be better if the authors strengthen the manuscript by answering the following questions;

1-    In evaluating the reliability of RWH systems. It is not very clear why the threshold volume value is 11.7 x105 m3. A scientific explanation is needed!

2-    The surface runoff that entered the dam reservoir at a rate of 5.2% of the total years, is it means throughout the years 1990-2020? Or in the maximum rainfall year? it is not clear.

3-    The authors mentioned in the abstract that “they applied a comprehensive software for everyday water balance using data from 20 years of daily rainfall to evaluate rainwater harvesting system (RWH) performance”. What kind of software? did they create this software or did they just apply it?

4-    Why the authors chose 1990-2020 as the basis for the analysis?

5-    There are some basic English correction issues in the text. proofreading is needed.

Author Response

  • Thanks for your efforts which identifying areas of our manuscript that needed modifications. The response to your comments is attached. 

Reviewer 2 Report (New Reviewer)

Comments-questions (to be clarified in the text):

1. Introduction: (a) What does GCM mean? (b) What is rcp?

2. See annotated manuscript! There are many points that should be reformulated.

Author Response

Thanks for your efforts which identifying areas of our manuscript that needed modifications. The response to your comments is attached. 

Round 2

Reviewer 2 Report (New Reviewer)

Comments - Questions (to be clarified in the text):

1. Lines 189 and 194: What is "macro"?

2. Lines 198-200: What does ID mean?

3. Line 281, Figure 3: According to Line 223, Rt refers to time-based reliability. Therefore, I suppose that TBR and Rt are identical terms. In this case, please, use the symbol Rt instead of TBR!

4. Figure 4: VR and Rv (volumetric reliability) are identical terms. In this case, please, use the symbol Rv instead of VR!

5. See annotated manuscript! 

Author Response

Thanks for your efforts which identifying areas of our manuscript that needed modifications. The response to your comments is attached. 

This manuscript is a resubmission of an earlier submission. The following is a list of the peer review reports and author responses from that submission.

Round 1

Reviewer 1 Report

 The article presents the rainwater catchment system reliability analysis for Al Abila Dam in Iraq's western desert. I have the following comments on the manuscript:

1.      The manuscript is not well organised. There is lack of integration between sentences throughout the manuscript. For example, I could not find any integration amongst the first three sentences in the Abstract.

2.      Specific problem of the research was not identified in the manuscript.

3.      Aim of the project is not well defined. In the abstract, it has been mentioned that extensive computer software was constructed. In the aim of introduction, construction of the software is not mentioned.

4.      Significance of the research is not provided in the introduction section. Novelty of the research was not identified in the introduction.

5.      The area of the catchment was mentioned 13,370 km2. Only one sampling point has been shown at the middle of the watershed area. If it is dam, how sampling point (dam) could be at the middle of the watershed area? How only one location could be the representative of the area?

6.      There is not integration between the aim and methods.

7.      The method was not adequately described. For example, storage capacity was shown in Figure 2. In the methods, what the storage categories used in the analysis is not mentioned.

8.      In the abstract, extensive computer software construction was mentioned. What software was developed? Which language was used?

9.      What is Imteaz equation?

10.   English of the manuscript is poor.

Reviewer 2 Report

The subject of the article is relevant to the aims and scope of Water journal. Research on alternative water sources is obviously necessary, especially in dry regions. Sustainable exploitation of natural water resources and their protection is of key importance for smart development. To achieve this, it is necessary to implement alternative sources of water such a rainwater and greywater. The major goal of this study was to evaluate rainwater harvesting systems reliability in a sub-catchment in Al-Anbar region in western Iraq.

In my opinion, such research is not novelty and has little interest in the international scale but the outcomes can be beneficial for designing local strategies of water management. There is a lot of research of rainwater harvesting systems.

In the Introduction, the authors provided a brief research background discussing the current state of modelling the rainwater harvesting systems. The Introduction is very general, it does not outline the research problem that the authors have dealt with. It does not show any novelty in relation to the already conducted research.

In the Methodology section there is no detailed information about the model. The authors refer to publication [19], but the main assumptions of the model and formulas should be described. There is also no detailed data entered into the model.

The Results and Discussion: the description of the research results is very poor, there is no comparison of the results and no discussion with the results of other authors. So I suggest that some discussion should be added to verify your results.

The text requires an editorial correction.